# G-Banding and Molecular Cytogenetics Detect Novel Translocations and Cryptic Aberrations in Human Immortal Endothelial Cells

**DOI:** 10.3390/ijms25147941

**Published:** 2024-07-20

**Authors:** Regina Lichti Binz, Rupak Pathak

**Affiliations:** Division of Radiation Health, Department of Pharmaceutical Sciences, College of Pharmacy, University of Arkansas for Medical Sciences, Little Rock, AR 72205, USA; rkbinz@uams.edu

**Keywords:** metaphase chromosome, endothelial biology, subtelomere, spectral karyotyping, florescence in situ hybridization

## Abstract

Endothelial cells (ECs) maintain vessel tone and barrier integrity, regulate blood homeostasis, and prevent the extravasation of leukocytes under normal physiological conditions. Because of the limited lifespans and batch-to-batch differences with respect to the genetic make-up of primary ECs, established immortal EC lines are extensively used for studying endothelial biology. To address this issue, the immortal endothelial cell line EA.hy926 was developed by fusing primary human umbilical vein endothelial cells (HUVECs) with human lung carcinoma A549 cells. EA.hy926 cells share a number of similar endothelial properties with HUVECs and are considered the immortal counterpart to primary HUVECs. However, the cytogenetic integrity of EA.hy926 cells is not fully elucidated. We characterized EA.hy926 cells with conventional G-banding and molecular cytogenetic techniques such as spectral karyotyping and subtelomeric fluorescence in situ hybridization. Cytogenetic analysis revealed an array of numerical and stable structural chromosomal rearrangements including one deletion, one duplication, one isochromosome, seven simple translocations, and five complex translocations in Ea.hy926 cells. These findings will advance comprehension of EA.hy926 cell biology and augment future endothelial studies, specifically in comparison studies between HUVECs and EA.hy926 cells.

## 1. Introduction

The endothelium (innermost layer of vasculature) is formed by a tightly packed monolayer of endothelial cells (ECs). ECs maintain vascular tone and barrier integrity, provide an antioxidant surface, regulate blood homeostasis, and prevent the attachment and extravasation of leukocytes [1,2,3]. Considering the best approach to elucidate endothelial biology, ex vivo primary cell culture systems are more advantageous than in vivo systems because ex vivo systems allow the streamlined manipulation of cells for the duration of an experiment and primary cells retain most of the physiological and functional characteristics of their in vivo counterparts.

Human umbilical vein ECs (HUVECs) presumably represent primary embryonic ECs, as they are derived from veins exclusive to the umbilical cord. In addition, HUVECs have the same modal chromosome number as post-embryonic ECs. Both HUVECs and adult ECs also form tubes, mediate angiogenesis and wound healing, and express similar molecular markers. However, because HUVECs are derived from the fetus as opposed to maternal tissue, biological sex affects their gene expression profile and functional activities. For example, there is significant variability in the response to thrombin by HUVECs derived from various umbilical cord sources [4]. HUVECs also cannot be grown in traditional culture medium; they require special endothelial culture medium, with a low serum level and various supplements. Additionally, primary ECs cannot be maintained ex vivo for an extended time. Therefore, immortalized EC lines, which can be cultured indefinitely and use traditional culture medium, are preferred for research.

The immortal hybrid HUVEC cell line—EA.hy926—was established by fusing HUVECs with human lung carcinoma A549 cells. EA.hy926 cells retain crucial endothelial properties [5,6]. Like primary HUVECs, EA.hy926 cells exhibit similar permeability, redistribution of tight junction proteins, and changes in structural proteins following thrombin treatment [4], as well as impairment of angiogenesis following high-glucose treatment in hypoxic conditions [6]. Moreover, HUVECs and EA.hy926 cells share a similar senescence phenotype following doxorubicin treatment [7]. However, EA.hy926 cells release tissue factor pathway inhibitors more readily for thrombin stimulation than HUVECs [8], while the senolytic agent ABT-263 (Navitoclax) selectively induces the apoptosis of DOX-induced senescent HUVECs but not in EA.hy926 cells [7], suggesting that the functional responses of EA.hy926 cells to various stimuli are diverse. The difference in cellular functions is closely associated with numerical and structural changes in the chromosomes. The current study elucidated the cytogenetic integrity of EA.hy926 cells, with the aim of assessing the difference in cytogenetic landscape between immortal EA.hy926 cells and primary HUVECs, which may help us to better understand the outcomes of comparative studies between these two cell lines. The conventional and molecular cytogenetic analyses revealed an array of numerical and structural cytogenetic alterations in EA.hy926 cells.

## 2. Results

### 2.1. G-Banding Revealed a Complex Karyogram for EA.hy926 Cells

We previously showed that HUVECs have a modal chromosome number of 46 [9], while the current study revealed a composite modal chromosome number of 75 for EA.hy926 cells based on the analysis of 15 metaphase spreads (1 with 72, 1 with 73, 3 with 74, and 10 with 75) (Appendix A). In a typical spread with 75 chromosomes, we observed one copy of chromosome 21; two copies of chromosomes X, 9, 13, 15, and 22; three copies of chromosomes 1, 2, 4, 5, 6, 7, 8, 10, 17, 18, and 19; four copies of chromosomes 3, 11, 12, 14, and 16; six copies of chromosome 20; and five marker chromosomes (i.e., with an unrecognizable banding pattern, which may or may not contain chromatin originating from various chromosomes) (Figure 1 and Appendix A). Out of 75 chromosomes in EA.hy926 cells, 54 chromosomes showed no change in banding pattern, while 21 chromosomes including 5 marker chromosomes displayed a change in banding pattern suggestive of rearrangements, duplications, or deletions.

### 2.2. SKY Identified Simple and Complex Translocations and Revealed Unique Clones in EA.hy926 Cells

Figure 2 shows a classified SKY image for an EA.hy926 cell with 75 chromosomes. SKY revealed 56 chromosomes with no change in spectral emission, 14 derivative chromosomes (der) resulting from simple translocations (t), and 5 der chromosomes resulting from complex translocations (Figure 2). Der chromosomes with a simple translocation include der(1)t(1;3), der(2)t(2;3), der(3)t(3;7), der(6)t(6;8), der(7)t(7;19), der(8)t(8;21), der(11)t(8;11), der(12)t(7;12), der(12)t(8;12) or der(12)t(12;21), der(14)t(1;14), der(14)t(8;14), der(14)t(14;21), der(19)t(15;19), and der(22)t(15;22). Chromosomes with complex translocations include der(1)t(8:2;1), der(3)t(3;2;5;2;11;17), der(6)t(6;1;9), der(8)t(8;X;1), and der(17)t(17;5;9).

The analysis of 50 metaphase spreads following SKY hybridization revealed two clones based on the status of a simple translocation involving a copy of chromosome 12 (one clone with chromosome 8 [26 out of 50] and the other with chromosome 21 [24 out of 50]), two clones based on one copy of chromosome X (one clone with chromosome 4 [9 out of 50] and the other without [41 out of 50]), and two clones based on one copy of chromosome 8p (one clone with no change [35 out of 50] and the other with a translocation from the chromatin of chromosome 10, 14, or 21 [15 out of 50]) (Appendix A).

### 2.3. Subtelomere FISH Identified Cryptic Aberrations in Chromosomes with No Change in Spectral Emission

Out of the 56 chromosomes with no change in spectral emission, 5 chromosomes exhibited an abnormal subtelomere hybridization pattern (Figure 3). One copy each of chromosomes X and 12 consistently showed no hybridization on the q subtelomere regions, indicating a deletion on each chromosome. Two copies of chromosome 16 showed consistent positive hybridization for tandem copies of the 16q subtelomere, indicating a duplication of the 16q subtelomeres. One copy of chromosome 18, which was initially identified as one of the marker chromosomes by G-banding, exhibited inconsistent positive hybridization for 18p subtelomere, suggesting the presence of two clones. One clone exhibited positive hybridization for 18p subtelomere and 9q subtelomere, defining a cryptic translocation [der(18)t(9;18)], and the other clone exhibited only 9q subtelomere, illustrating the formation of a ring chromosome. One copy of chromosome 19 showed a positive hybridization for 19p13 and 19p subtelomere on both sides of the centromere, indicating isochromosome 19.

### 2.4. Subtelomere FISH Probes Defined the Origins of Telomeric Regions of Chromosomes Involved in Simple Translocations

Figure 4 shows the distribution of subtelomeres in the 14 derivative (der) chromosomes resulting from simple translocations, showing their respective G-banded chromosomes with corresponding classified SKY and subtelomere FISH images. We observed positive hybridization for 1p and 3q subtelomeres for der(1)t(1;3); 2p and 3q subtelomeres for der(2)t(2;3); 3p and 7q subtelomeres for der(3)t(3;7); 8q and 6q subtelomeres for der(6)t(6;8); 7p and 19q subtelomeres for der(7)t(7;19); 8q and 21q subtelomeres for der(8)t(8;21); 11p and 8q subtelomeres for der(11)t(8;11); 7p and 12q subtelomeres for der(12)t(7;12); 12p and 8q subtelomeres for der(12)t(8;12) or 12p and 21q subtelomeres for der(12)t(12;21); 1p and 14q subtelomeres for der(14)t(8;14); 4p, 8p, and 14q subtelomeres for der(14)t(4;8;14); 21q and 14q subtelomeres for der(14)t(14;21); 15q and 19q subtelomeres for der(19)t(15;19); and 15q and 22q subtelomeres for der(22)t(15;22). Notably, the der(14)t(4;8;14) revealed positive interstitial hybridization of the 8p subtelomere probe (immediately distal to the chromosome 14 centromere), while the 4p subtelomere probe showed positive hybridization at the terminal end of the p arm.

### 2.5. Subelomere FISH Defined the Origins of the Telomeric Regions of Chromosomes Involved in Complex Translocations

Figure 5 shows the distribution of subtelomere regions for the 14 derivative (der) chromosomes resulting from simple translocations, as well as their respective G-banded chromosomes with corresponding classified SKY and subtelomere FISH images. We observed positive hybridization for 8p and 1q subtelomeres for der(1)t(8:2;1), 3p and 17q subtelomeres for der(3)t(3;2;5;2;11;17), 6p and 9q subtelomeres for der(6)t(6;1;9), 8p and 1q subtelomeres for der(8)t(8;X;1), and 17p and 9q subtelomeres for der(17)t(17;5;9). Notably, 6q and Xq probes showed interstitial hybridization on der(6)t(6;1;9) and der(8)t(8;X;1), respectively.

## 3. Discussion

EA.hy926 cells are considered an immortal counterpart of human primary endothelial cells: HUVECs. Despite exhibiting similarities in endothelial functions, primary HUVECs and immortal EA.hy926 cells show different responses to the same stimuli [7,8,10]. Difference in cellular response to a specific stimulus is tightly associated with the level and functional activity of the proteins involved in the specific signaling pathways, which can be altered because of numerical and/or structural changes in chromosomes. Therefore, to understand the difference in the cytogenetic landscapes between two cell lines, the cytogenetic characterization of EA.hy926 cells is crucial.

HUVECs, like all other human adult somatic cells, have a modal chromosome number of 46 (diploid) involving no simple or complex translocation but with the deletion of the Xq terminal region in 50% cells, as previously described by our group [9]. The current study demonstrated that the modal chromosome number of EA.hy926 cells is 75, which is considered near triploid and significantly higher than the modal chromosome number of its parental cell lines: HUVECs and A549. A previous study also reported that EA.hy926 cells have a higher modal chromosome number than the HUVECs and A549 [11], which corroborates our current data. However, the authors reported that the modal chromosome number of EA.hy926 cells is 80 [11], which is comparable with our current findings. This marginal discrepancy in modal chromosome number in EA.hy926 cells between two studies could be the result of differences in passage number and/or cell culture conditions. Interestingly, the same study reported that immediately after establishing the EA.hy926 cell line, the modal chromosome number was 100, which subsequently decreased to 80 after a few passages [11]. These data clearly suggest that over time, cell lines may undergo considerable cytogenetic alterations.

In terms of chromosomal structural abnormalities, a previous study reported that EA.hy926 cells contain a dicentric-like derivative chromosome [11]. We also observed similar dicentric-like chromosomes in EA.hy926 cells. Considering that dicentric chromosomes are unstable—structural abnormalities and therefore, cannot persist through subsequent passages—we can be certain that this chromosome is not dicentric based on our in-depth study using SKY and subtelomeric FISH probes. Further, this chromosome is involved in a complex translocation involving chromosomes 6, 1, and 9 with 6p and 9q subtelomeres at both ends of the chromosome, with positive interstitial hybridization of the 6q subtelomere (Figure 5). In addition, we identified 4 complex aberrations and 14 simple translocations, which were previously unreported. Interestingly, we recognized two simple translocations, der(11)t(8;11) and der(19)t(15;19), which were previously identified in A549 cells by another group [12]. Careful comparison of our findings with those of the previous study of A549 revealed common breakpoints on chromosomes 1q, 2q, 3q, and 12q [12]. We have not investigated the functional implications of these translocations; however, translocations are associated with the pathogenesis of several diseases including cancer. Importantly, numerical and/or structural chromosomal instability is considered the hallmark feature of cancer pathogenesis [13,14]. Thus, lending clinical relevance to our current findings.

Finally, we positively identified two X chromosomes by SKY. FISH confirmed positive hybridization of two chromosome X centromeres. Subtelomere FISH further revealed positive hybridization on only one Xq homologue, which corroborates our previous findings [9]. Because HUVECs are one of the parental cell types of EA.hy926 cells, it can be assumed that the copy of the X chromosome with the deletion in the Xq terminal region originated from HUVECs; however, further validation of the presence of Xq terminal regions in other parental cell types (human lung carcinoma A549 cells) is needed to validate this claim. Previous studies indicated that EA.hy926 cells are different in terms of the biological behavior and expression of proteins compared to their parental cell types: HUVECs and A549 [10,12,15]. The current study provides a potential explanation that the aforementioned changes could be due to the altered cytogenetic landscape of EA.hy926 cells compared to HUVECs and A549, which may augment future comparative analyses.

## 4. Materials and Methods

### 4.1. Cell Culture

EA.hy926 cells (ATCC) were cultured in Dulbecco’s Modified Eagle’s Medium (ATCC) supplemented with 10% fetal bovine serum (ATCC) and 1% antibiotics. Cell cultures were maintained in a humidified incubator with 5% CO_2_ at 37 °C and subcultured every 2 to 3 days with a brief 0.25% trypsin–EDTA (Gibco, MA, USA) treatment.

### 4.2. Metaphase Chromosome Preparation

Metaphase chromosomes were prepared as previously described [9,16,17,18]. Cells were released with trypsin and pelleted by centrifugation following mitotic arrest with KaryoMAX colcemid (75 ng/mL) (Gibco, Waltham, MA, USA). Cell pellet was resuspended in hypotonic solution (75 mM KCl; Gibco) at 37 °C, treated with fixative (3:1 Methanol: Glacial acetic acid), pelleted by centrifugation, and resuspended with fresh fixative for a total of 3 fixative washes. Dilute cell suspension was applied to clean slides at room temperature (RT).

### 4.3. Conventional G-Banding

G-banded chromosomes were prepared as described elsewhere [9,16,17,18]. Slides were aged by baking overnight (67 °C), treated with 0.025% trypsin working solution at RT, rinsed twice in Tyrode’s buffer solution (Sigma, St. Louis, MO, USA), and stained in 4% Giemsa working solution (Sigma) for 5 min. The band level of resolution for karyotypes was determined according to the International System for Human Cytogenomic Nomenclature (2020) [19]. At least 15 G-banded metaphases were analyzed for each cell type.

### 4.4. Spectral Karyotyping (SKY)

Probe cocktail and concentrated antibodies were obtained from Applied Spectral Imaging (ASI) and used as described elsewhere [9,16,17,18]. In brief, SKY probe and slides were denatured separately. The denatured probe was applied to slide, protected with a glass coverslip, and sealed with rubber cement. Slides were incubated in a humidified chamber for 48 h at 37 °C and washed in formamide wash solution (50% formamide in 2× SSC) 3 times. Blocking reagent, Cy5 antibody solution, and Cy5.5 antibody solution were sequentially applied, incubated, and washed. Slides were counterstained with DAPI. SKY images were captured using an SD200 Spectracube (ASI, Carlsbad, CA, USA) mounted on a Zeiss Imager.Z2 microscope (Dublin, CA, USA). At least 50 SKY images were captured under 63× magnification for spectral analysis for each cell type.

### 4.5. FISH Analysis

The ToTelVysion kit (Vysis Inc./Abbott Molecular Laboratories, Abbott Park, IL, USA) consists of 15-probe cocktails to demark the subtelomere regions of all human chromosomes. The FISH technique and a description of the probe cocktails have been provided elsewhere [9]. In brief, probe and slides were denatured separately. Probe mixtures were applied to the respective target areas of cells, protected with a coverslip, and sealed with rubber cement. Slides were incubated in a humidified chamber overnight at 37 °C. Post-hybridization wash was performed at 45 °C. Slides were counterstained with DAPI and protected with a coverslip. At least 4 cells from each hybridization area were photographed and analyzed, for a total of 60 cells (GenASIs software, version GV80, ASI, Carlsbad, CA, USA).

## 5. Conclusions

In conclusion, to the best of our knowledge, we identified an array of simple and complex chromosomal aberrations that had not previously been reported. These findings will augment our knowledge of the EA.hy926 karyotype, improve our ability to design mechanistic studies or biomedical and pharmaceutical research on EA.hy926 cells, and provide critical information to apply to future endothelial biology studies. Based on our current findings, we propose the following composite nomenclature for EA.hy926 cells: 72~75<3n>,X,der(X)add(X)(q27),−X,del(1)(q31q31),der(1)t(8;2;1)(pter→p12;q22.3→q33; q10→qter),der(1;3)t(1;3)(p10;q10),der(2)t(2;3)(p10;q?26),der(3;7)t(3;7)(p10;q10),+der(3) t(3;2;5;2;11;17)(pter→q25;q~22.3→q?;?q→?q;~q22.3→q?;~q21→q22;q12→qter),der(6)del(6) t(6;8)(p11;q21.2), psu dic(6)t(6;1;9)(q27;q11→q25.3;~q31),der(7)t(7;19)(q11.1;q13.3),der(8) t(8;21)(p11.1;q11.1),der(8)t(8;X,1)(q12;q28→~q22;q32.1),−9,+der(11)del(11)t(8;11)(q24.1; q23),add(12)(q24.3),+der(12)t(7;12)(q35;p12),+der(12)del(12)t(8;12)(q24.1;~q14),−13,der(14)t(1;14)(p12;p11),+der(14)t(4;8;14)(p16.2;p11.2→p23;p11),der(14;21)t(14;21)(q10;q10), −15,+16,dup(16)(q24q24)x2,+der(17)t(17;5;9)(q21;q?→q?;~q31),+der(18)t(8;18)(~q24.1;q22),der(18)del(18)t(9;18)(q34;q11.2),der(19)t(19;19)(p12;p10),+der(19)t(15;19)(q22;p11),+20, +20,+20,?add(22)(p11),der(22)t(15;22)(~q24;p11)[cp26]/72~75,idem,−der(12)del(12)t(8;12) (q24.1;~q14),+der(12)del(12)t(12;21)(~q14;q22.3)[cp24].

## Figures and Tables

**Figure 1 ijms-25-07941-f001:**
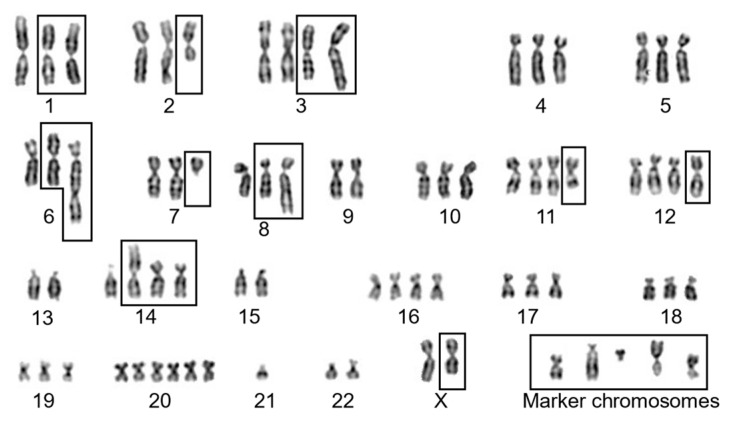
G-banded karyogram for EA.hy 296 cells. In total, 54 chromosomes show an apparently normal banding pattern, while 21 chromosomes display abnormal banding patterns (including 5 marker chromosomes) indicative of rearrangement, duplication, or deletions (as indicated by black boxes). The numbers listed below indicate chromosome number, except for marker chromosomes, as detected by G-banding.

**Figure 2 ijms-25-07941-f002:**
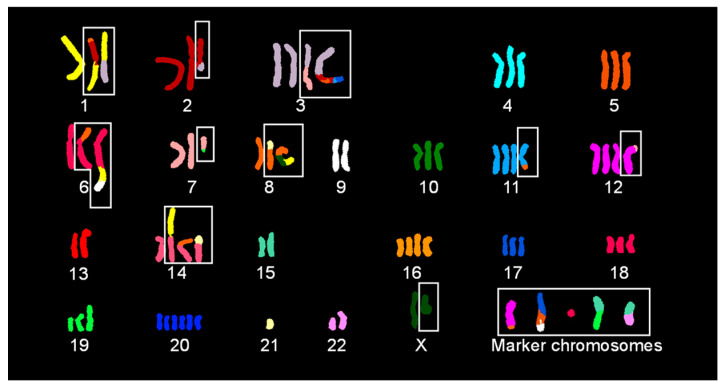
Classified image of spectral karyotyping of the EA.hy926 cell. Out of 75 chromosomes, 56 chromosomes showed no change in spectral emission, 14 derivative chromosomes each resulting from a simple translocation, and 5 derivative chromosomes each resulting from complex translocations (as indicated by white boxes). The numbers listed below indicate chromosome number, except for marker chromosomes, as detected by spectral karyotyping and each color represents a particular chromosome.

**Figure 3 ijms-25-07941-f003:**
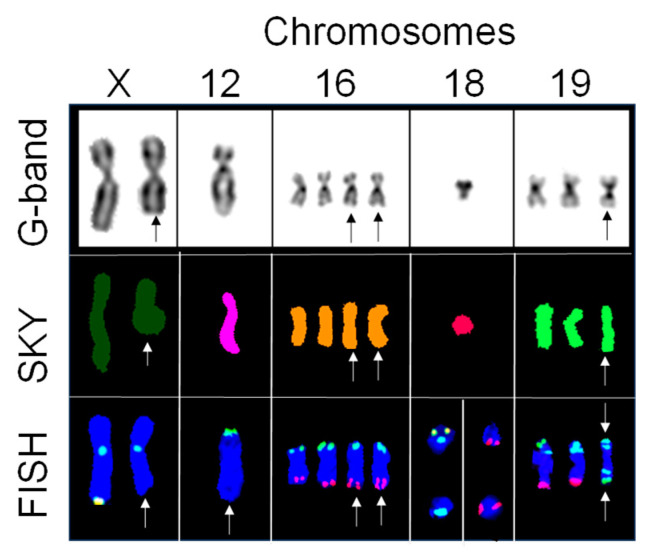
Cryptic translocations identified with subtelomere FISH in chromosomes that showed no change in spectral emission by SKY. Partial karyograms of G-band, SKY classified images, and subtelomere FISH for chromosomes X, 12, 16, 18, and 19 counterstained with DAPI (blue). Chromosome X exhibits positive hybridization for X centromere (aqua) and positive hybridization for only one homologue of Xq (yellow), thus confirming a deletion in one homologue (indicated by arrow). Chromosome 12 shows positive hybridization for 12p subtelomere (green), while negative hybridization for the 12q subtelomere regions suggests a deletion (indicated by arrow). Chromosome 16 exhibits positive hybridization for 16p subtelomeres (green) and 16q subtelomeres (red), confirming the microduplication (indicated by arrows) of the terminal q arms for two copies of chromosome 16. Chromosome 18 shows the positive hybridization of 18p subtelomere (yellow) and 18 centromere (aqua) and positive hybridization of the 9q subtelomere (red), confirming a cryptic translocation of the 9q subtelomere with the marker chromosome 18(31/60). Negative hybridization of 18p subtelomere, positive hybridization of the 18 centromere (aqua), and positive hybridization of the 9q subtelomere (red), confirming a cryptic ring involving the 9q subtelomere with the marker chromosome 18 (17/60). Chromosome 19 shows positive hybridization for 19p subtelomere (green), 19p13 (aqua), and 19q subtelomere (red), with a signal pattern consistent with two normal chromosomes 19 and one isochromosome 19.

**Figure 4 ijms-25-07941-f004:**
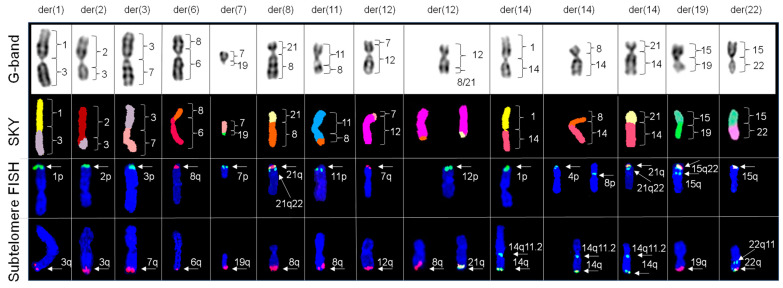
Distribution and origin of telomeric regions identified with subtelomere FISH in chromosomes identified by SKY to show simple translocations. Partial karyograms of G-bands, SKY classified images, and subtelomere FISH images counterstained with DAPI (blue) for derivative (der) chromosomes 1, 2, 3, 6, 7, 8, 11; two unique copies of 12; and three unique copies of 14, 19, and 22. One copy of der 1 exhibits positive hybridization for 1p (green) and 3q (red) subtelomeres. One copy of der 2 exhibits positive hybridization for 2p (green) and 3q (red) subtelomeres. One copy of der 3 exhibits positive hybridization for 3p (green) and 7q (red) subtelomeres. One copy of der 6 exhibits positive hybridization for 8q (red) and 6q (red) subtelomeres. One copy of der 7 exhibits positive hybridization for 7p (green) and 19q (red) subtelomeres. One copy of der 8 exhibits positive hybridization for 21q (yellow), 21q22 (aqua),and 8q (red) subtelomeres. One copy of der 11 exhibits positive hybridization for 11p (green) and 8q (red) subtelomeres. One copy of der 12 exhibits positive hybridization for 7q (red) and 12q (red) subtelomeres. Another copy of der 12 exhibits positive hybridization for 12p (green) and 8q (red) or 21q (yellow) subtelomeres. One copy of der 14 exhibits positive hybridization for 1p (green), 14q11.2 (aqua), and 14q (yellow) subtelomeres. Another copy of der 14 exhibits positive hybridization (left) for 4p (green), with positive interstitial hybridization for 8p (green) subtelomere (right), 14q11.2 (aqua), and 14q (yellow). Another copy of der 14 exhibits positive hybridization for 21q (yellow), 21q22 (aqua), and 14q (yellow) subtelomeres. One copy of der 19 exhibits positive hybridization for 15q (yellow), 15q22 (aqua) and 19q (red) subtelomeres. One copy of der 22 exhibits positive hybridization for 15q (yellow), 22q11 (aqua) and 22q (yellow) subtelomeres.

**Figure 5 ijms-25-07941-f005:**
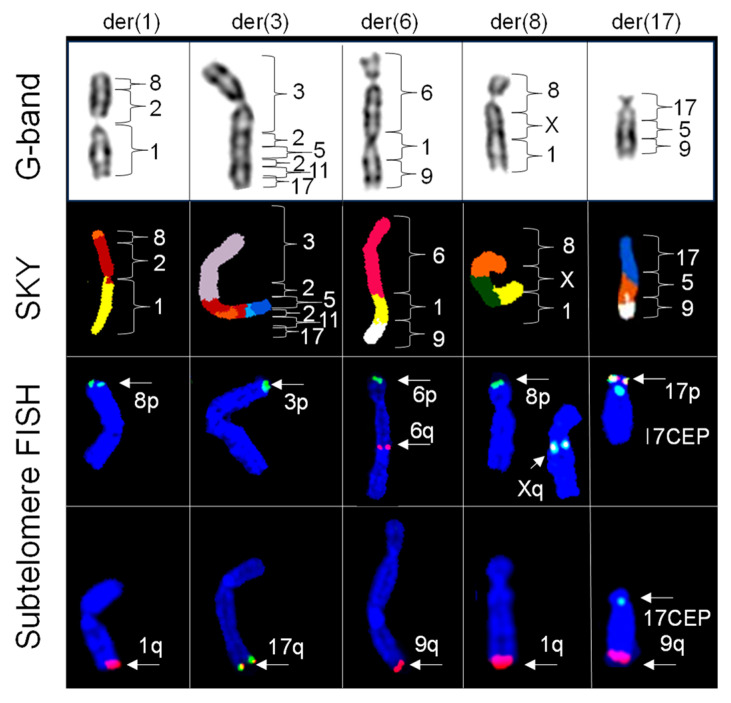
Distribution and origins of telomeric regions identified with subtelomere FISH in chromosomes with complex translocations identified by SKY. Partial karyograms of G-bands, SKY classified images, and subtelomere FISH images counterstained with DAPI (blue) for derivative (der) chromosomes 1, 3, 6, 8, and 17. One copy of der 1 exhibits positive hybridization for 8p (green) and 1q (red) subtelomeres. One copy of der 3 exhibits positive hybridization for 3p (green) and 17q (yellow) subtelomeres. One copy of der 6 exhibits positive hybridization for 6p (green), interstitial 6q (red) and 9q (red) subtelomeres. One copy of der 8 exhibits positive hybridization for 8p (green) interstitial Xq (yellow) and 1q (red) subtelomeres. One copy of der 17 exhibits positive hybridization for 17p (yellow), 17 CEP (aqua) and 9q (red) subtelomeres.

## Data Availability

The original contributions presented in the study are included in the article/Appendix A, further inquiries can be directed to the corresponding author.

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
