# Peer review of "G-Banding and Molecular Cytogenetics Detect Novel Translocations and Cryptic Aberrations in Human Immortal Endothelial Cells"

_ijms, 2024, doi:10.3390/ijms25147941_

Round 1

Reviewer 1 Report

Comments and Suggestions for Authors

In manuscript ijms-3083186 the authors report a comprehensive cytogenetic characterization of cell line EA.hy926, a line derived by the fusion between primary human umbilical vein endothelial cells (HUVECs) and human lung carcinoma A549 cells. This cell line was created to obtain an immortal endothelial cell line. To date, an extensive cytogenetic description of this line is not present in literature. Overall, the work is interesting, but I have some doubts about its use in the scientific community, for the following reason. In the abstract, the authors affirm that “Cytogenetic analysis revealed an array of numerical and stable structural chromosomal rearrangements”. However, they also affirm that in another study (reference 11, lines 196-203) the same line started with a modal chromosome number of ca. 100, to rapidly reach a modal number of ca. 80, while they find a modal number of 75. In addition, the authors of ijms-3083186 also report the presence of some chromosomes missing telomeric regions (figure 3 and lines 126-129), a situation that is intrinsically unstable as it is prone to end-to-end fusions (either between sister chromatids or among non-homologous chromosomes) and potentially bringing to the formation of dicentric chromosomes. Moreover, also a ring chromosome (lines 133-136) is described in one clone. All together, these data seem far to be suitable to describe a “stable” cell line, thus potentially impairing the results they obtained. The authors should comment this in the Discussion and justify why they are positive about considering this cytogenetic description useful in the long term.

Additional minor points.

1.      In lines 27-28 the verb is missing.

2.      Line 84: “supplemenal” is misspelled.

3.      In Table S1 the range is 72-76 but no line in the table reaches a value of 76. Why?

Author Response

Reviewer Comment: In manuscript ijms-3083186 the authors report a comprehensive cytogenetic characterization of cell line EA.hy926, a line derived by the fusion between primary human umbilical vein endothelial cells (HUVECs) and human lung carcinoma A549 cells. This cell line was created to obtain an immortal endothelial cell line. To date, an extensive cytogenetic description of this line is not present in literature. Overall, the work is interesting, but I have some doubts about its use in the scientific community, for the following reason.

Response: We appreciate that reviewer recognized no other comprehensive literature is available and finds our work interesting.

Reviewer Comment: In the abstract, the authors affirm that “Cytogenetic analysis revealed an array of numerical and stable structural chromosomal rearrangements”. However, they also affirm that in another study (reference 11, lines 196-203) the same line started with a modal chromosome number of ca. 100, to rapidly reach a modal number of ca. 80, while they find a modal number of 75.

Response: Thank you for noticing the inconsistency of reported modal number for EA.hy926. Careful review of the reference manuscripts has improved confidence of our results. Previous cytogenetic studies on EA.hy926 were performed using only conventional staining methods immediately after establishing the EA.hy926 cell line in the year of 1983. The authors refer to the modal number in a nonspecific manner, reporting the modal number to be “about 80”, which is close to the modal number we reported in the current study. Further, there is neither photographic evidence nor a table to illustrate evidence for distribution of chromosomes. Considering that we performed extensive analysis using both conventional (15 cells, supplemental table 1) and molecular cytogenetic methods (50 cells), with a total of 65 cells karyotyped, we have provided strong evidence that 75 is an accurate representation for the modal number of EA.hy926 cells obtained from ATCC.

Reviewer Comment: In addition, the authors of ijms-3083186 also report the presence of some chromosomes missing telomeric regions (figure 3 and lines 126-129), a situation that is intrinsically unstable as it is prone to end-to-end fusions (either between sister chromatids or among non-homologous chromosomes) and potentially bringing to the formation of dicentric chromosomes. Moreover, also a ring chromosome (lines 133-136) is described in one clone. All together, these data seem far to be suitable to describe a “stable” cell line, thus potentially impairing the results they obtained.

The authors should comment this in the Discussion and justify why they are positive about considering this cytogenetic description useful in the long term.

Response: Thank you for this astute observation. We agree with this comment and have edited the manuscript to reflect this perspective more accurately. Loss of the entire telomere region would, indeed result in chromosomal instability. We have reported only 2 chromosomes lacking positive hybridization for subtelomere probes. This could be the result of either an interstitial deletion of only the subtelomere region, or a cryptic translocation of satellite material originating from one of the acrocentric chromosomes, therefore, the chromosomes with apparently missing sub-telomeric region are not necessarily involved in structural aberrations, thus supporting the establishment of a stable cell line. Moreover, cytogenetic characterization was performed on EA.hy926 cells various passages and observed the same chromosome landscape, further supporting cytogenetic stability of this cell line.

Additional minor points.

Reviewer comment: In lines 27-28 the verb is missing.

Response: Thank you, this issue has been resolved.

Reviewer comment: Line 84: “supplemenal” is misspelled.

Response: Thank you, this issue has been resolved.

Reviewer comment:  In Table S1 the range is 72-76 but no line in the table reaches a value of 76. Why?

Response: Thank you, this typographical error has been corrected.

Reviewer 2 Report

Comments and Suggestions for Authors

Authors did an obvious and nonetheless necessary study in EA.hy926 cells to characterize its genetic basics. 

Major comments:

- authors did no telomere-FISH - they did FISH uding subtelomeric probes. Telomere FISH uses 6bp repeat probes being identical for all telomeres - please fix that throughout text. Thanks.  

- authors need to write their results as karyotype formula, like e.g. 72~75<3n>,X,der(X),der(1)t(1;3),der(1)t(1;6),der(2)t(2;3),der(3)t(3;7),+der(3)t(3;2;5;2;11;17)(3pter->3q25::2q?22.3->2q?::5?q->5?q::2q?22.3->q?::11q?->11q22::17q12->17qter), etc. They need to check their formula with ISCN(2020); yet the formual is not correct.

- Authors need to align their results with those of cell line A549 as published e.g. in PMID: 20545000.

- please comment on the suitability of such a 'mixed cellline' from two sources for research.

Author Response

Reviewer Comment: authors did no telomere-FISH - they did FISH using subtelomere probes. Telomere FISH uses 6bp repeat probes being identical for all telomeres - please fix that throughout text. Thanks.  

Response: Thank you for bringing this valid point to our attention. Not only does this comment improve the clarity of our results, but it also allows us to provide a better response to reviewer-1 concerns. Our manuscript has been edited accordingly.

Reviewer Comment: authors need to write their results as karyotype formula, like e.g. 72~75<3n>,X,der(X),der(1)t(1;3),der(1)t(1;6),der(2)t(2;3),der(3)t(3;7),+der(3)t(3;2;5;2;11;17)(3pter->3q25::2q?22.3->2q?::5?q->5?q::2q?22.3->q?::11q?->11q22::17q12->17qter), etc. They need to check their formula with ISCN(2020); yet the formual is not correct.

Response: Thank you for this suggestion. We have carefully reviewed the International System for Human Cytogenomic Nomenclature (ISCN) (2020), which allowed correction for subtle errors. The new nomenclature is as follows:

72~75<3n>,X,der(X)add(X)(q27),-X,del(1)(q31q31),der(1)t(8;2;1)(pterèp12;q22.3èq33;q10èqter),der(1;3)t(1;3)(p10;q10),der(2)t(2;3)(p10;q?26),der(3;7)t(3;7)(p10;q10),+der(3)t(3;2;5;2;11;17)(pterèq25;q~22.3èq?;?qè?q;~q22.3èq?;~q21èq22;q12èqter),der(6)del(6)t(6;8)(p11;q21.2),                                                                                                         psudic(6)t(6;1;9)(q27;q11èq25.3;?31),der(7)t(7;19)(q11.1;q13.3),der(8)t(8;21)(p11.1;q11.1),der(8)t(8,X,1)(q12,q28èq?22;q32.1),-9,+der(11)del(11)t(8;11)(q24.1;q23),add(12)(q24.3),+der(12)t(                                                                                                        7;12)(q35;p12),+der(12)del(12)t(8;12)(q24.1;q?14),-13,der(14)t(1;14)(p12;p11),+der(14)t(4;8;14)(p16.2;p11.2èp23;p11),der(14;21)t(14;21)(q10;q10),15,+16,dup(16)(q24q24)x2,+der(17)t(17;5;9)(q21;q?èq?;q?31),+der(18)t(8;18)(q?24.1;q22),der(18)del(18)t(9;18)(q34;q11.2),der(19)t(19;19)(p12;p10),+der(19)t(15;19)(q22;p11),+20,+20,+20,?add(22)(p11),der(22)t(15;22)(q?24;p11)[cp26]/ 72~75,idem,der(12)del(12)t(8;12)(q24.1;q?14),+der(12)del(12)t(12;21)(q?14;q22.3)[cp24]

In addition, we have edited our results to reflect a more detailed representation of cytogenetic nomenclature based on the following information.

From Chapter 3 (p 34)

When more than one symbol or abbreviation is used together, a space is placed between the two.

From section 4.1 General principles (p 38)

In the description of a karyotype the first item to be recorded is the total number of chromosomes, including the sex chromosomes, followed by a comma (,). The sex chromosome constitution is given next.

72~75,<3n>,X,der(X)add(X)(q27),-X

In order to distinguish homologous chromosomes, one of the numerals may be underlined (page 39)

For example: in the nomenclature for EA.hy926, the underlined information specifies that each piece of chromosome 1 originated from the same homologue in spite of their various sequence in the nomenclature.  del(1)(q31,q31),… der(1;3)t(1;3)(p10;q10),… psu dic(6)t(6;1;9)(q27;q11èq25.3;?31)… der(8)t(8,X,1)(q12,q28èq?22;q32.1)…

And

der(19)t(19;19)(p12;p10),+der(19)t(15;19)(q22;p11),…the underlined information specifies that each piece of chromosome 19 originated from the same homologue. In this case the t(15;19) is mentioned AFTER the t(19;19) as it represents a gain of the entire chromosome.

From section 4.3.1.3 Four-break and more complex rearrangements (p 43)

Whenever applicable, the guidelines for three-break rearrangements should be used. Unbalanced rearrangements will lead to at least one derivative chromosome and in these situations the use of the symbol der to describe the derivative chromosome(s) is recommended. It will usually not be possible to adequately describe all complex rearrangements with the short form. The detailed form can always be used to describe any abnormality, however complex. Still, it may be necessary to illustrate the rearrangement and/or describe it in words to ensure complete clarity.

4.3.2.1 Additional symbols (p 44)

A single colon (:) is used to indicate a chromosome break and a double colon (::) to indicate break and reunion. In order to avoid an unwieldy description, and arrow (è or ->), meaning from – to, is employed. The end of a chromosome arm may be designated either by its band designation or by the symbol ter (terminal)…

der(1)t(8;2;1)(pterèp12;q22.3èq33,q10èqter)

der(14)t(4;8;14)(p16.2;p11.2èp23;p11)

4.4 Derivative chromosomes (p 44)

A derivative chromosome is a structurally rearranged chromosome generated by (1) more than one rearrangement within a single chromosome, e.g., and inversion and a deletion of the same chromosome, or (2) rearrangements involving two or more chromosomes, e.g., the unbalanced product(s) of a translocation. An abnormal chromosome in which no part can be identified is referred to as a marker chromosome.

Derivative chromosomes are designated der. The term always refers to the chromosome(s) that has an intact centromere or neocentromere. The derivative chromosome is specified in parentheses, followed by all aberrations involved in the generation of the derivative chromosome. The aberrations should be listed according to the breakpoints of the derivative chromosome from pter to qter and should not be separated by a comma.

From page 45…The full karyotype designation needs be written only once in any given publication and then can be abbreviated (i.e., breakpoints not included).

From section 5.1 Questionable identification

From page 48. A question mark (?) indicates questionable identification of a chromosome or chromosome structure. It is placed either before the uncertain item, or it may replace a chromosome, region, or band designation.

For example: in der(2)t(2;3)(p10;q?26), the question mark is placed to indicate the certainty of the q arm, but the uncertainty of the specific band/locus.

From section 5.2. (p 49) Uncertain Breakpoint localization or chromosome number

A tilde(~) is used to denote intervals and to express uncertainty about breakpoint localizations in that it indicated the boundaries of a chromosome segment in which the breaks may have occurred.

From Chapter 9 (p 68) A pseudodicentric chromosome is a dicentric structure in which only one centromere is active Such chromosomes are abbreviated psu dic, and the segment with the presumptively active centromere, based on the morphology in the majority of cells, is always written first.

psu dic(6)t(6;1;9)(q27;q11èq25.3;?31)

From section 11.2 Modal number (p 94)

The modal number is the most common chromosome number in a tumor cell population. The modal number may be expressed as a range between tow chromosome numbers

Response: Optional wording

The authors recognize the omission of description of chromosome X.

Cytogenetic nomenclature is written based on correct chromosome placement—which is based on the origin of the centromere.

Once the chromosomes are correctly placed based on the centromere of origin, the nomenclature is written such that the reader can interpret the information for each chromosome in the order for which it appears in the karyogram. The exception for this is the sex chromosomes, which should come immediately following the modal number.

In circumstances of whole arm translocations and the analyst is unable to determine the true origin of the centromere, the chromosome is placed in the karyotype based on the lowest number chromosome, with the proper orientation of the chromosome of interest. Using der(1;3)t(1;3)(p10;q10) reflects that the analyst recognizes a whole arm translocation and cannot identify with which chromosome the centromere originates.

Reviewer Comment: Authors need to align their results with those of cell line A549 as published e.g. in PMID: 20545000.

Response: Thank you for bringing this manuscript to our attention. We have carefully reviewed it to compare findings with our study. The limitations from this study are (1) the karyotype is only presented in the form of conventional G-banding, and (2) a molecular study was performed, but the images are not easy to discern. In spite of the aforementioned limitations, we were able to recognize some obvious similarities and have highlighted these in our discussion section in the revised version and also added the reference (please see reference # 12).

Reviewer Comment: please comment on the suitability of such a 'mixed cellline' from two sources for research.

Response: Our study helps illustrate an important aspect—the chromosomal rearrangements present in EA.hy926 are stable. We have subcultured, harvested, and analyzed chromosomes at various passages and observed the same chromosome landscape. This cell line is extensively used to research endothelial cell biology. The information gleaned from our study will serve as an important reference for those who use these cells for comparative studies with primary endothelial cells, specifically HUVECs or A549. Because this cell line is considered as immortal counterpart of HUVECs and used extensively for comparative studies yet there is no cytogenetic study that is presented with the comprehensive detail of our current study.

Round 2

Reviewer 1 Report

Comments and Suggestions for Authors

No additional comments/corrections are needed.

Author Response

Thank you so much for positive recommendation.

Regards

Rupak

Reviewer 2 Report

Comments and Suggestions for Authors

Authors did almost all revisions requested

just 2 things: 

- Please correct in line 284 

International System for Human Cytogenetic Nomenclature

to 

International System for Human Cytogenomic Nomenclature (2020)

and add actual reference 

- Title must read as "Banding and molecular cytogenetic...

 conventional cytogenetics means just Giemsa stain and not banding studies as done here

Thanks

besides paper is ok from my side now

Author Response

We have changed the title of our manuscript as suggested by the reviewer and also corrected ISCN 2020 and provided the reference, please see reference number 19.